# Integrated miRNA–mRNA Analysis Reveals Critical miRNAs and Targets in Diet-Induced Obesity-Related Glomerulopathy

**DOI:** 10.3390/ijms25126437

**Published:** 2024-06-11

**Authors:** Marina López-Martínez, Maria Pilar Armengol, Irina Pey, Xavier Farré, Paula Rodríguez-Martínez, Mireia Ferrer, Esteban Porrini, Sergio Luis-Lima, Laura Díaz-Martín, Ana Elena Rodríguez-Rodríguez, Coriolano Cruz-Perera, Marta Alcalde, Maruja Navarro-Díaz

**Affiliations:** 1CSUR National Unit of Expertise for Complex Glomerular Diseases of Spain, Nephrology Department, Vall d’Hebron University Hospital, Vall d’Hebron Institute of Research, 08035 Barcelona, Spain; 2Department of Medicine, Universitat Autònoma de Barcelona, Bellaterra, 08913 Barcelona, Spain; 3Genomic Platform, Germans Trias i Pujol’s Research Institute, Badalona, 08916 Barcelona, Spain; 4Pathology Department, Germans Trias i Pujol Hospital, Badalona, 08916 Barcelona, Spain; 5Statistics and Bioinformatics Unit, Vall d’Hebron Research Institute, 08035 Barcelona, Spain; 6Laboratory of Renal Function (LFR), Faculty of Medicine, University of La Laguna, Complejo Hospitalario Universitario de Canarias, 38320 La Laguna, Spainlauradiazmart@gmail.com (L.D.-M.);; 7Instituto de Tecnologías Biomédicas (ITB), Faculty of Medicine, University of La Laguna, La Laguna, 38320 Tenerife, Spain; 8Department of Laboratory Medicine, Complejo Hospitalario Universitario de Canarias, La Laguna, 38320 Tenerife, Spain; 9Research Unit, Hospital Universitario de Canarias, La Laguna, 38320 Tenerife, Spain; 10Fundación General de la Universidad, University of La Laguna,38320 Tenerife, Spain; 11Comparative Medicine and Bioimage Centre of Catalonia (CMCiB), Fundació Institut d’Investigació en Ciències de la Salut Germans Trias i Pujol, Badalona, 08916 Barcelona, Spain; 12Pharmaco and Device Epidemiology Group, CSM, NDORMS, University of Oxford, Oxford OX1 3PT, UK; 13Nephrology Department, Sant Joan Despí Moisès Broggi Hospital, Sant Joan Despí, 08970 Barcelona, Spain

**Keywords:** microRNA, biomarker, targetome, obesity-related glomerulopathy, mesangial matrix increase, podocyte hypertrophy, Wistar rats

## Abstract

This study aimed to investigate obesity-related glomerulopathy (ORG) at cellular, structural, and transcriptomic levels. Thirty Wistar rats were randomized into two groups: 15 rats were fed with a standard diet (SD-rats), and 15 rats were fed with a high-fat diet (HFD-rats). After 10 weeks, the weight, kidney function, histological features, and transcriptomic changes were assessed. HFD-rats gained significantly more weight (55.8% vs. 29.2%; *p* < 0.001) and albuminuria (10,384.04 ng/mL vs. 5845.45 ng/mL; *p* < 0.001) compared to SD-rats. HFD-rats exhibited early stages of ORG, with predominant mesangial matrix increase and podocyte hypertrophy (PH). These lesions correlated with differentially expressed (DE) genes and miRNAs. Functional analysis showed that miR-205, which was DE in both the kidneys and urine of HFD-rats, negatively regulated the PTEN gene, promoting lipid endocytosis in podocytes. The downregulation of PTEN was proved through a higher PTEN/nephrin ratio in the SD-rats and the presence of lipid vacuoles in HFD-podocytes. This study has found a specific targetome of miRNAs and gene expression in early stages of ORG. Also, it emphasizes the potential value of miR-205 as a urinary biomarker for detecting podocyte injury in ORG, offering a tool for early diagnosis, and opening new avenues for future therapeutic research of obesity-related glomerulopathy.

## 1. Introduction

Obesity is a modifiable risk factor for the development and progression of chronic kidney disease (CKD) [1,2]. The risk of new-onset albuminuria, one of the two criteria documented for CKD diagnosis, increases by 51% in individuals with obesity [1]. In 2017, 697.5 million cases of all-stage CKD were recorded [3], an increase that was partially explained by the growing prevalence of obesity worldwide. Regrettably, 2.6 million deaths in 2017 were attributable to impaired kidney function, indicating that all-age CKD mortality increased by 41.5% between 1990 and 2017 [3]. The spread of obesity and its associated comorbidities has led to global health crises.

The pathogenesis of obesity-related glomerulopathy (ORG) is unclear, as not all individuals with obesity develop CKD. Clinical manifestations of ORG include glomerular hyperfiltration, proteinuria, and renal impairment, making it a secondary form of focal segmental glomerulosclerosis (FSGS) [4]. Interestingly, before secondary FSGS is established, drastic weight loss through bariatric surgery reverses the initial clinical alterations in ORG [5]. Therefore, an early diagnosis should be mandatory to prevent the development and progression of CKD.

In a previous study, we first described the presence of early-stage ORG in patients with severe obesity without overt clinical renal manifestations [6]. Therefore, many individuals with obesity may be at risk for developing chronic kidney disease (CKD) without definitive clinical evidence for detection. This poses a challenge in identifying kidney damage at a stage where it could still be potentially reversible. Moreover, in cases where patients exhibit only microalbuminuria, the ethical considerations around the necessity of a kidney biopsy arise, especially when specific treatments for obesity-related glomerulopathy (ORG) are lacking. The need for antiproteinuric treatment remains consistent in such scenarios [7]. Undoubtedly, the discovery of non-invasive biomarkers for ORG would enable early diagnosis without subjecting patients to unnecessary risks.

To deepen our knowledge of the preliminary phase of ORG, further preclinical research into the cellular and molecular changes is of special interest. At the molecular level, these processes are regulated by the integrated actions of various damaging and protective/regenerative biological pathways. Scientific investigation of epigenetic modifications is currently in the pipeline not only for the characterization of diseases but also for prognosis and therapeutic approaches [8]. So-called “microRNAs” (miRNAs) are short, non-coding RNAs (ncRNAs) that play principal roles in normal physiological processes as well as in the development of diseases [9]. For example, miRNAs play critical roles in modulating gene expressions implicated in cell differentiation, growth/proliferation, migration, metabolism, and defense and exhibit tissue-specific patterns [10].

The field where miRNA analysis has been extensively studied is cancer. Indeed, consistent data suggest that the dysregulation of specific ncRNAs contributes to cancer development by upregulating or silencing target genes. Epigenetic signatures have been demonstrated to be characteristic of many cancers, and miRNAs also appear to be independent prognostic factors of the disease [8,10]. Also, some miRNAs have been identified as being involved in kidney diseases. Moreover, some miRNAs have been evaluated, in clinical trials, as therapeutic targets, such as in Alport syndrome and autosomal dominant polycystic kidney disease. Intriguingly, miRNA expression profiling studies have found that miRNAs also influence the initiation and progression of obesity-related kidney injury [11]. Considered as plausible non-invasive biomarkers, some urinary exosomal microRNAs in obesity-associated CKD have already been isolated [11]. Identifying miRNAs implicated in the early stages of ORG in the urine of patients with obesity could allow forward diagnosis and treatment. Furthermore, the stratification of patients into low- and high-risk categories for developing CKD would reduce the overall disease burden [11].

Overall, understanding the initial changes in obesity-related renal disease at the cellular, structural, and transcriptomic levels will help to recognize the pathogenic phenomena that occur in this disease. To enhance our knowledge of ORG and identify potential biomarkers, this study aimed to establish the first interactome of miRNAs, their targetome, and the presence of kidney lesions in obesity.

## 2. Results

### 2.1. Weight Gain in Rats with Diet-Induced Obesity

The SD-fed rats reached a median weight of 461.33 ± 37.36 g (29.20% gain) after 10 weeks, whereas the HFD-fed rats gained 55.83% (544.06 ± 43.66 g), *p* < 0. 001 (Figure 1). Although all 30 rats developed obesity, significant differences in absolute weight were observed between the two groups (*p* < 0.001, Student’s *t*-test). The HFD-fed rats developed obesity between weeks 2 and 6, whereas half the SD-fed rats developed obesity by week 10. The rate of weight gain in the HFD group was 2.3 times higher than that in the SD group.

### 2.2. Renal Function

Albuminuria levels were measured in 24 h urine samples collected and correlated with the body weight increase (*r* = 0.702; *p* < 0.001). The HFD-fed rats exhibited higher albuminuria levels than the SD-fed rats (10,384.04 ± 1167.96 ng/mL vs. 5845.45 ± 2616.79 ng/mL; *p* < 0.001, respectively). Renal function, assessed by the iohexol plasma clearance technique, was not significantly different between the two groups (67.8 ± 11.6 mL/min in the SD-fed rats vs. 71.67 ± 10.8 mL/min in the HFD-fed rats).

### 2.3. Study of Renal Morphological Changes

The rat kidneys of both the SD and HFD groups were surrounded by adipose tissue, with the kidneys of the HFD-fed rats appearing larger than those of the SD-fed rats. All the HFD-fed rats exhibited early ORG lesions observed in humans (mesangial matrix increase (MMI), podocyte hypertrophy (PH), mesangial cell proliferation (MCP), and glomerulomegaly), whereas the SD-fed rats only exhibited MMI and PH (Figure 2a–f). Morphometric analysis revealed a higher MMI in the HFD group than in the SD group (a mean of 93% of the glomeruli analyzed vs. 53%; *p* < 0.001). The glomerular area was significantly larger in the HFD group than in the SD group (10,640.92 ± 1220.62 μm^2^ vs. 8562.28 ± 838.10 μm^2^; *p* = 0.003). Enlarged and vacuolated proximal tubular cells with signs of protein reabsorption were also observed (Figure 2g). No focal or segmental glomerulosclerosis, interstitial fibrosis, or tubular atrophy was observed. TEM revealed swollen podocytes with an increased number of cytoplasmic organelles, especially mitochondria and lipid vacuoles (Figure 2h–j). The HFD group had a lower podocyte density (18.09%) than the SD group (23.65%) (Figure 3).

### 2.4. Diet-Induced Transcriptomic Changes and Validation of DE RNAs in Rat Kidneys

To profile diet-related gene changes in the kidneys, RNA-Seq experiments were performed to compare gene expression levels in the SD (*n* = 9) and HFD (*n* = 8) groups, detecting 14,700 expressed genes. Hierarchical clustering revealed similarities between the samples from the SD and HFD groups, although they were not entirely distinct (Figure 4a). Genes were considered as DE if the adjusted *p*-value was <0.05 and the absolute log2FoldChange was >1, yielding four clearly DE genes (Epha3 and Atp12a were upregulated; Robo3 and Sult4a1 were downregulated in the HFD group). Additionally, 11 genes showed a tendency toward DE, including Htr5b, Ccl20, and Pros1 (Figure 4b). These genes are implicated in renal injury prevention (Epha3 and Pros1), inflammation (CCL20), and nervous system function (Sult4a1, Robo3, and Htr5b). Then, qPCR confirmed that ATP12a, along with Epha3 (*p* = 0.00278) and Pros1 (*p* = 0.00831), was five-times upregulated in the HFD group (*p* = 0.00130). Conversely, Sult4a1 and Htr5b were downregulated in the HFD group (*p* = 0.00230 and *p* = 0.01924, respectively), whereas Robo3 showed no significant difference (*p* = 0.43209) owing to high variability (Figure 4c).

### 2.5. Lesion-Related Transcriptomic Changes and DE RNAs in Rat Kidneys

MMI and PH were the most common histological lesions observed in the rat model. To understand the role of gene expression in lesion types, we analyzed transcriptomic data based on the presence of MMI and PH, regardless of the diet. Only glutathione S-transferase alpha 3 (Gsta3) is downregulated in rats with PH, increasing their vulnerability to oxidative stress and contributing to renal interstitial fibrosis [12]. Gsta3 regulates the epithelial–mesenchymal transition and is crucial for renal fibrosis. Conversely, X Kell blood group precursor-related family member 4 was upregulated in the same group and implicated in apoptotic process engulfment [13]. Conserved oligomeric Golgi complex subunit 4 (Cog4) and phytoplankton are upregulated in rats with MMI. Cog4 [14] is expressed in tubules and is linked to kidney clear cell carcinoma, whereas PHYHIPL [15] is enriched after ischemic brain injury (Figure 4d,e).

### 2.6. Small RNA Profile Changes According to Diet and Weight

Unsupervised hierarchical clustering was used to identify potential groups based on molecular expression profiles. This analysis revealed a slight, but not substantial, distinction between the HFD and SD control rats. A total of 812 smallRNA sequences were detected, including mitochondrial tRNA (Mt tRNA) (*n* = 22), ribosomal RNA (rRNA) (*n* = 36), small Cajal body-specific RNA (scaRNA) (*n* = 14), small nucleolar RNA (snoRNA) (*n* = 312), small nuclear RNA (snRNA) (*n* = 90), and microRNA (miRNA) (*n* = 338). Fifteen smallRNA sequences met the criteria of an adjusted *p*-value of <0.05 and an absolute log2FoldChange of >1, with most being downregulated in HFD-fed animals (Figure 5a). The miRNAs were identified using miRTarBase and TargetScan, and nine of these were differentially expressed (DE): miR-205, miR-140-5p, miR-185-5p, miR-128-3p, miR-28-5p, miR-340-3p, miR-6329, miR-674-3p, and miR-148b-5p (Figure 5b,c). GO analysis revealed targets associated with inflammatory responses, tissue injury and repair, and functions related to the nervous system, fat metabolism, and mitochondria. Quantitative PCR (qPCR) confirmed that miR-205 was upregulated, while miR-6329 and miR-340-3p were downregulated in HFD-fed rats (Student’s *t*-test, *p* = 0.0170, *p* = 0.0490, and *p* = 0.0453, respectively). Other miRNAs showed trends toward differential expression but lacked statistical significance because of the high variability among the samples (Figure 5d).

### 2.7. Functional Analysis of Pathways and Leading-Edge Genes Targeted by DE miRNAs

To identify the DE miRNAs’ targetome, we intersected, predicted, and validated miRNA targets with mRNAs showing negative correlations (Spearman’s correlation coefficients of <−0.5) with the miRNAs. We identified 712 putative gene targets for the 11 DE miRNAs: 601 genes related to one miRNA, 30 genes related to two miRNAs, and 1 gene (PTEN) related to 4 DE miRNAs (miR-22-3p, miR-22-5p, miR-144-3p, and miR-205) (Figure 5e). Additionally, 79 genes were not expressed in the rat kidneys. Given the complex relationships between miRNAs and their targets, we focused on the combined upregulation of miR-22-3p, miR-22-5p, miR-144-3p, and miR-205, which negatively regulate the PTEN gene function (Appendix A). To confirm the transcriptomic data, immunofluorescence staining was performed on the kidneys of the SD- and HFD-fed rats. These data revealed that the SD-fed rats had a higher PTEN–nephrin ratio than the HFD-fed rats, confirming the downregulation of PTEN in the HFD-fed rats (Figure 6).

### 2.8. Lesion-Related Transcriptomic Changes in Rat Kidneys (small RNA)

To understand the relationship among mRNA expression, detected miRNAs, and histological lesions (mesangial matrix increase (MMI) and podocyte hypertrophy (PH)), we conducted in silico miRNA functional analyses using DIANA-miRPath v3.0. We analyzed the expression data of miRNAs across all the rats, identifying significant associations with MMI and PH based on a *p*-value of <0.05 and an absolute log2FoldChange of >1. For MMI, 75 genes, including Nr2f6, Dtx2, Ptpn23, Paqr6, Fam193b, Potpnm1, Mier2, Phf1, Slc22a17, Nfkbil1, Ankrd9, and Rexo1, met both criteria. Similarly, for PH, 54 genes, including Adm, Ugt8, Cd24, Tsnaxip1, Thegl, Pdxk, Hsp4, Socs3, Ptpn23, Junb, Mfsd12, and Slc22a17, met the criteria (Figure 7).

To address the possibility of chance correlations, we used simulations with 100,000 random datasets to compare the observed gene expressions with randomly generated datasets. This helped us to ensure that the observed correlations between gene expressions and histological lesions were statistically significant and not due to random chance. The average expression of each gene, with or without lesions, was calculated relative to the overall expression, confirming the robustness of our findings.

Immunofluorescence staining was performed to partially validate the transcriptomic data. T-cells were mainly present in the interstitial zone of the HFD group, with scattered CD3+ cells observed in the peritubular capillaries and some glomeruli. No T-cells were observed in the SD group, and no CD20+ B-cells were detected in either group. Additionally, M1 macrophages (CD86+ cells) were identified only in the HFD-fed rats.

### 2.9. Diet-Dependent Expressions of Identified Small RNAs in Rat Urine Samples

We analyzed urinary miRNAs to analyze the kidney pathology and injury sites, mirroring smallRNA analysis in rat kidneys by mapping the Rnor6.0 genome’s detected 471 sequences (240 Mt_tRNA, 46 rRNA, 2 scaRNA, 32 snoRNA, 45 snRNA, and 106 miRNA sequences). Among them, 53 were DEs (adjusted *p*-value < 0.1; absolute log2FoldChange > 1) in SD-fed vs. HFD-fed rats; 18 were downregulated in HFD-fed rats, and 35 were downregulated in SD-fed rats (Figure 8). Interestingly, miR-205 was the only overlap between the kidney and urine samples. All the DE miRNAs were upregulated in the HFD-fed rats. Twelve miRNAs that correlated with urine weight, including miR-101b-3p, miR-30a-5p, and miR-205, were significantly upregulated in HFD-fed rats. Other miRNAs showed trends but lacked statistical significance owing to their high variability (Figure 9).

### 2.10. Lesion-Related Transcriptomic Changes in Urine

Urinary data, such as those from the kidneys, were analyzed based on the presence of MMI or PH. Lesion-related results had a *p*-value of <0.05 and an absolute log2FoldChange of >1. No miRNA–tissue injury association was observed with MMI, although four miRNAs (miR-1843a-5p, miR-21-5p, miR-125b-5p, and miR-29a-3p) distinguished rats with PH (Figure 10).

## 3. Discussion

This study established a rat model of diet-induced obesity with early-stage ORG. The paired mRNAs and miRNAs were sequenced. Wistar rats with diet-induced obesity developed glomerulomegaly, MMI, PH, MCP, and intracytoplasmic lipid deposits in podocytes. Our findings describe the immunological, regeneration, and injury processes that occur in this disease when the damage could be reversed by treatment. We also identified a robust signature of miRNAs, in the urine, that could be explored as a non-invasive biomarker and potential therapeutic target for ORG.

The HFD-fed rats gained more weight, showing greater MMI and larger glomeruli, than the SD-fed rats. The most frequently encountered lesion was MMI, whereas MCP and podocyte intracytoplasmic lipid deposits were observed only in the HFD group. The role of lipid accumulation in non-adipose tissue in the development of ORG has attracted increasing interest [16]. Podocytes also slightly increased in cytoplasmic organelles, especially mitochondria. Abnormal mitochondrial structures in the obesity milieu cause the downregulation of the key fatty acid oxidative enzymes and upregulation of enzymes involved in lipogenesis [17,18]. The HFD-fed rats experienced greater podocyte depletion (Figure 3) and albuminuria, which were almost twice greater than those experienced by the SD-fed rats, indicating early ORG progression [19,20]. The GFR measurements showed no significant intergroup differences, suggesting similar hyperfiltration levels despite weight gain [21].

The adiporenal crosstalk suggests chronic subclinical inflammation in the adipose tissue [22,23]. Alterations in lipid metabolism trigger macrophage migration, inhibitory factor expression, and the release of inflammatory activators, such as CD40 and interleukin-6 [16]. In the HFD-fed group, T-cells were present in the interstitial zone, and CD3+ cells were in the peritubular capillaries and glomeruli. Certainly, T-cells have also been described as being implicated in obesity–inflammation and metabolic disease [24]. M1-type macrophages promote proinflammatory cytokine production, whereas M2-type macrophages synthesize anti-inflammatory cytokines and aid in tissue repair [4]. Obesity prompts a shift from M2 to M1 macrophages in the adipose tissue, exacerbating inflammation [22]. In fact, experimental approaches suggest that the inhibition of M1 macrophages may reduce kidney injury [25]. Accordingly, we identified CD86-marked cells, indicating the presence of pro-inflammatory M1 macrophages, in HFD-fed rats.

A comparison of the gene expression profiles in HFD- and SD-fed rats revealed four gene groups with differential expressions as follows (Figure 4c): ATP12a, related to potassium–proton exchange ATPase activity; Epha3 and Pros1, involved in renal injury prevention and tissue protection; CCL20, linked to inflammation and T-cell recruitment [26]; and Sult4a1, Robo3, and Htr5b, associated with the nervous system [27,28,29]. CCL20 hyper-expression in the proximal tubular epithelial cells of HFD-fed rats likely recruits immune cells, particularly M1 macrophages, thereby intensifying inflammation. To counteract this, M2 macrophages, Pros1, and Epha3 aid in tissue regeneration to maintain kidney function. Pros1 mediates cytoprotection in diabetic kidneys by inhibiting glomerular endothelial cell and podocyte apoptosis [30].

Notably, miRNAs are small non-coding RNA molecules that play a critical role in regulating gene expression. They are involved in various biological processes and are essential for normal organ development and homeostasis, including kidney development and function. Their dysregulation has been implicated in the development and progression of ORG [31]. Of special interest was miR-205, the only miRNA that was upregulated in the HFD-fed rats; miR-205 is involved in the epithelial-to-mesenchymal transition when podocytes lose the expression of highly specialized protein markers and acquire mesenchymal markers. It has been described as a prognostic marker for tubulointerstitial damage in patients with IgA nephropathy [32]. Concerning miRNAs downregulated in the HFD-fed rats, miR-140-5p affects adipocyte differentiation [33] and mediates renal fibrosis [34]; miR-185-5p ameliorates endoplasmic reticulum stress and renal fibrosis by downregulating ATF6 [35]; miR-128-3p regulates adipogenesis and lipolysis [36] and promotes the acute kidney injury process by activating the JAK2/STAT1 pathway [37]; miR-28 is a negative regulator of the germinal center reaction and regulates B-cell receptor signaling, proliferation, and apoptosis [38]; and miR-340-3p is downregulated in the skeletal muscle of mice after HFD feeding [39]. Interestingly, miR-6329 is associated with hypertension in rats [40], and miR-674-3p is associated with acute kidney injury [41].

The relationships between the DE miRNAs and their target genes were also analyzed. Approximately 700 genes were associated with these miRNAs. Specifically, the upregulation of miR-205 and its interactions with miR-22-3p, miR-22-5p, and miR-144-3p inhibited exclusively a single gene, PTEN (Figure 5e). The effect of this inhibition on the HFD-fed rats was confirmed through the higher PTEN/nephrin ratio in the SD-fed group (Figure 6). In podocytes, PTEN protects the kidneys by enhancing autophagy, preventing cell death, and maintaining podocyte integrity. In a diet-induced obesity model using C57BL/6J mice, Ji et al. explored miR-141-3p’s role in mitochondrial dysfunction in hepatic cells by inhibiting PTEN [42]. Other authors have suggested that PTEN downregulation increases endoplasmic reticulum stress in a study conducted with podocyte cell cultures [43]. A previous study on mice with a podocyte-specific PTEN knockout showed the typical features of ORG when fed a high-fat diet [44]. This suggests that the loss of the PTEN function contributes to ORG development. PTEN downregulation is associated with increased lipid endocytosis in podocytes, leading to lipid accumulation and the aggravation of ORG [44]. Indeed, in the present study, the podocyte cytoplasm of the HFD group showed empty spaces compatible with lipid vacuoles (Figure 2i). To the best of our knowledge, this study is the first to describe PTEN downregulation in a model of diet-induced obesity-related ORG in non-genetically modified rodents. Insights into PTEN’s roles in podocyte lipid endocytosis and glomerulopathy may offer novel therapeutic approaches for kidney diseases characterized by lipid accumulation and metabolic disturbances, such as obesity.

In kidney diseases, urine may offer a significant advantage over tissues as a non-invasive biomarker owing to its direct access to damaged tissue, particularly in the early stages of the disease. In the current animal model, we observed a differential gene expression in kidney tissue based on the presence of MMI or PH (Figure 4d,e and Figure 7). In addition, there were DE miRNAs in the urine, depending on the presence of PH (Figure 10). Strikingly, the individual evaluation of the DE miRNAs in the urine revealed that the same DE miRNA in the kidney tissue (miR-205) of the HFD-fed rats was also statistically DE in the HFD-fed rats compared with DE in the SD-fed rats (Figure 9b). Consequently, these findings could contribute to the detection of urinary biomarkers for PH or podocyte injury in patients with obesity, aiming to prevent ORG progression to CKD.

This study has limitations. The limited sample size may have compromised the generalizability of the findings. Moreover, translating the results from animal models to humans requires caution and validation in clinical studies. Nevertheless, this was the first exploratory project, and further experimental and clinical studies will be conducted in the future. The main strength of this study is that by integrating innovative techniques, molecular biology, histopathology, and clinical data analysis, it bridges these fields of research and offers a multidimensional understanding of ORG.

In conclusion, miRNAs may play a role in ORG. Specifically, the upregulation of miR-205 in the kidneys and its detection in urine are associated with podocyte lipid endocytosis, which occurs in the early stages of ORG through the inhibition of the PTEN gene. Thus, miR-205 could serve as a plausible urinary biomarker for early-stage ORG and pave the way for future therapeutic studies. Moreover, the analysis of urinary smallRNAs identified DE genes, depending on whether the rats developed PH. Future studies are needed to confirm the involvement of these miRNAs in human ORG.

## 4. Materials and Methods

### 4.1. Establishment of the Obesity-Induced Animal Model

A total of 30 male Wistar Han rats (300–350 g; Charles River Laboratories España S.A.), aged 7 weeks and not genetically modified, were housed and fed under standard laboratory conditions. They were randomized into two groups: control rats (*n* = 15) fed a standard diet (SD) (TEKLAD 2014; percentages by weight: 14.3% protein, 4% fat, 18.0% neutral detergent fiber, and 2.9 Kcal/g) and rats (*n* = 15) fed a high-fat diet (HFD) (TEKLAD 06414; percentages by weight: 23.5% protein, 34.4% fat, 27.3% carbohydrate, and 5.1 Kcal/g) (Harlan Laboratories, Indianapolis, IN, USA) for 10 weeks. The weight and food intake were measured weekly. The rats were considered as being obese if their weight gain was >25% of their initial baseline weight. They were housed in metabolic cages for 24 h before organ collection to prevent the urine from being contaminated with feces or feed.

### 4.2. Organ and Urine Harvesting

The kidneys were harvested from each animal and stored in a saline solution at 4 °C to remove excess adipose tissue. A quarter of each renal tissue was fixed in formaldehyde and embedded in paraffin for immunochemistry and histological studies; another quarter was frozen in an isopentane bath and preserved at −80 °C. The remaining tissue was saturated in an RNA-later stabilization solution (Thermo Fisher Scientific Inc., Waltham, MA, USA) and stored at 4 °C for gene expression determination (16 of 30 biopsies for transcriptomic and morphological studies; 14 for quantitative polymerase chain reaction experiments). Urine from 16 rats was centrifuged at 2000× *g* for 10 min, and aliquots were stored at −80 °C.

### 4.3. Measurement of Renal Function: Albuminuria and Iohexol Clearance

Albuminuria was assessed via colorimetric enzyme-linked immunoassays (Abcam, Cambridge, UK) with a sensitivity of 0.44 ng/mL and concentrations of 0.195–50 ng/mL. A total of 10 µL of urine diluted 1:100 with the appropriate dilution buffer (including standards) was added to the wells and precoated and blocked with a rat-albumin-specific antibody. Subsequently, an anti-albumin biotinylated detection antibody and streptavidin–peroxidase conjugate were added, followed by visualization using 3,3′,5,5′-tetramethylbenzidine. The samples were processed in triplicate.

The glomerular filtration rate (GFR) was measured through the plasma clearance of iohexol [45]. After slight sedation with isoflurane (2.5%), 200 μL of Omnipaque 300 solution (GE Healthcare, Milan, Italy) containing 129.4 mg of iohexol was injected intravenously into one lateral tail vein of conscious rats that were placed in a plexiglass rat restrainer. Blood samples (approximately 15 μL) were collected before and 20, 40, 80, 120, and 140 min after injection. Samples were mixed by gentle vortexing for 7 min, collected again using heparinized capillary tubes (10 μL), deposited on filter paper (Whatman 903, GE Healthcare, Cardiff, UK), and allowed to dry for at least 24 h. Then, iohexol plasma concentrations in the total blood samples were measured by high-performance liquid chromatography–ultraviolet spectroscopy using a C18 reverse-phase column (Advanced Chromatography Technologies Ltd., Scotland) in a high-performance liquid chromatography system (Agilent Series 1260, Santa Clara, CA, USA) equipped with a diode array detector set at 254 nm. Plasma concentrations of iohexol were recalculated from blood levels using the formula Cplasma1/4 Cblood/1—hematocrit. The hematocrit was determined using the formula hematocrit 1/4 (H1/H2) 100, where H1 is the height of the red blood cell column, and H2 is the height of the red blood cell column plus the height of the plasma column after centrifuging a heparinized capillary tube filled with blood for 10 min at 6000 rpm. The reproducibility of the plasma clearance of iohexol in rats using the method described was <10%.

### 4.4. Morphometry, Histology, Immunohistochemistry, Immunofluorescence, and Transmission Electron Microscopy (TEM)

Standard histopathology techniques were applied for 3 µm thick rat kidney sections. The glomeruli, particularly those sectioned through the hilium, were morphometrically analyzed for area and mesangial expansion. Thirty glomeruli per sample were assessed in periodic-acid–Schiff- and hematoxylin-and-eosin-stained sections. Slides were scanned at a 40 µm resolution using a PANNORAMIC^®^ 1000 Flash DX scanner and analyzed using QuPath (v0.4.4) [46]. Mesangial matrix increase was defined as increased mesangial extracellular material with an interspatial width of >2 mesangial cell nuclei in one or more peripheral mesangial areas. Mesangial cell proliferation was defined as the presence of more than three mesangial cells surrounded by the mesangial matrix in an intact glomerular segment in 3 µm thick sections. Podocyte hypertrophy was defined as podocyte enlargement with large nuclei and prominent nucleoli, with or without intracytoplasmic protein resorption droplets.

Immunohistochemical staining (BenchMark ULTRA automated IHC/ISH slide staining system; Roche Ventana, Tucson, Arizona) was performed using antibodies against WT1 (clone 6F-H2, mouse monoclonal antibody; Ventana) and Ki67 (clone 30-9, rabbit monoclonal antibody; Ventana).

The immunohistochemical slides were scanned and assessed similarly with histochemical techniques.

The immunocytochemistry involved blocking the peroxidase activity with 3% H_2_O_2_ in methanol for 5–15 min (Thermo Fisher Scientific). Anti-CD3 and anti-CD19 antibodies (Thermo Fisher Scientific) were used to distinguish between T- and B-cells, with goat anti-mouse F(ab)2 used as the secondary antibody. Sections were stained with alkaline phosphatase (Abcam, Cambridge, UK), counterstained with hematoxylin, and observed under a Zeiss Axioskop-2 microscope.

Double immunofluorescence was performed for PTEN–nephrin and macrophage M1–M2 staining using two combinations of rabbit anti-PTEN (Abcam), mouse anti-nephrin (Santa Cruz, CA, USA), mouse anti-CD86 for M1 macrophages (Abcam), and rabbit anti-CD206 for M2-type macrophages (Thermo Fisher). Goat anti-mouse A488 and anti-rabbit Cy3 secondary antibodies were used as received. The nuclei were stained with 4′,6-diamidino-2-phenylindole (Thermo Fisher Scientific). Slides were viewed using an Abberior STEDYCON confocal microscope, and ImageJ software was used to analyze the mean fluorescence of all the markers.

TEM was used to analyze six rat samples (three in the SD group and three in the HFD group), as per the standard protocol. Tissues from the paraffin blocks were dewaxed, hydrated, post-fixed, dehydrated, and embedded in epoxy resin. Ultrathin sections were cut using a Leica Ultracut microtome and stained with uranyl acetate and lead citrate using a Leica EM stainer. Samples were examined using a JEM 1010 TEM (JEOL USA, Inc., Peabody, MA, USA) at 80 kV, and digital images were captured using an Orius CCD camera (Gatan Inc., Pleasanton, CA, USA).

### 4.5. Total RNA Isolation and miRNA Enrichment

RNA from 30 rat kidneys (15 SD- and 15 HFD-fed rats) was extracted using TRIzol reagent (Invitrogen, Thermo Fisher). DNase I treatment (Invitrogen) and quality control (QC) analysis were performed using a TapeStation 2200 (Agilent). Half the RNA was subjected to RNA-Seq, and the remainder was used for quantitative polymerase chain reaction (qPCR).

Then, smallRNAs from 30 rat kidneys were purified using 1 μg of the total RNA with SPRIselect beads (Bee Na Lee, Beckman Coulter, Brea, CA, USA). Sixteen kidneys were used for smallRNA-Seq and 14 for qPCR validation. Sixteen rat urine samples were collected, and 500 μL of the samples was used for smallRNA isolation with the ZR urine RNA isolation kit (Zymo Research Corp, Irvine, CA, USA). QC was assessed using a bioanalyzer with a high-sensitivity chip (Agilent).

### 4.6. Library Preparation, RNA Sequencing, and Data Analysis

Sixteen RNA-Seq libraries were prepared from 1 μg of the total RNA using an NEBNext^®^ Ultra™ II directional RNA library prep kit for Illumina^®^ (New England Biolabs, NEB, Ipswich, MA, USA). The HighSeq 2500 system generated 40–50 M reads/sample. FastQC was used to assess the read quality, and Cutadapt was used to remove adapters. The Salmon tool was used to map and quantify transcripts to the Rnor6.0 genome. DESeq2 v1.24.0 was used for differential expression analysis.

### 4.7. Library Preparation, smallRNA Sequencing, and Data Analysis

Sixteen smallRNA libraries were prepared using the NebNext smallRNA library prep set (Illumina, San Diego, CA, USA). Fragments were size-selected on a 6% polyacrylamide gel (Invitrogen), and library quality control (QC) was conducted with a TapeStation 2200 using HiSense D1000 screen tape (Agilent). The HighSeq 2500 system (Illumina) generated 12–15 M reads/sample. FastQC was used to assess the read quality, and Cutadapt was used to remove adapters. The Subread/Rsubread package was used to map and quantify transcripts in the Rnor6.0 genome. Annotations were sourced from different databases based on smallRNA classes (http://www.mirbase.org/ftp.shtml (accessed on 1 February 2023) https://www.ensembl.org/info/data/ftp/index.html (accessed on 1 February 2023). QCs included a multidimensional scaling plot analysis and Euclidean distances between samples to ensure the suitability of the normalized data. DESeq2 v1.24.0 Bioconductor’s package was used for differential expression testing. The miRNA target sites were cataloged based on experimental validation and computational prediction using the multiMiR Bioconductor package (http://multimir.ucdenver.edu) (accessed on 1 February 2023). The miRNAs with an adjusted *p*-value of ≤0.05 and a Log2FoldChange of >1 were deemed as being upregulated; a Log2FoldChange of <−1 indicated downregulation.

### 4.8. Predicted and Validated Targetome for miRNAs

It is known that miRNAs regulate gene expression by promoting degradation or repressing the translation of target transcripts; miRNA target sites have been cataloged in databases (miRecords, miRTarBase, and TarBase for validated targets and DIANA-microT, ElMMo, MicroCosm, miRanda, miRDB, PicTar, PITA, and TargetScan for predicted targets) based on experimental validation and computational prediction using a variety of algorithms. In this study, multiMiR Bioconductor’s package [47] (http://multimir.ucdenver.edu) was used for this purpose. To categorize the validated gene targets, the keywords were grouped into five categories as follows: inflammatory response (including adhesion, leukocytes, chemokines, and adipokines); kidney damage (extracellular matrix and renal injury molecules); mitochondria (metabolism and respiratory chain); adipose tissue; and nerve-related tissue. A computer algorithm searched the Entrez summary of each gene for these keywords and classified the genes accordingly.

### 4.9. Gene ontology and Pathway Analyses

The miRNA gene ontology (GO) and pathway analyses utilized DIANA-miRPath v3.0 [47], examining 14 differentially expressed (DE) miRNAs with validated target genes across the groups. DIANA-TarBase v7.0 [48] was used to investigate enriched pathways using Fisher’s exact test meta-analysis with Benjamini–Hochberg’s false discovery rate (FDR) correction (*q*-value < 0.05). Functional enrichment analysis employs pathways from the Kyoto Encyclopedia of Genes and Genomes (KEGG) database after the statistical analysis to identify pathways significantly targeted by the selected miRNAs. Heat maps facilitate the identification of miRNA subclasses or definition of pathways.

### 4.10. Reverse Transcription and Quantitative Reverse Transcription Polymerase Chain Reaction Validation

The total RNA extracted from the rat kidneys underwent retrotranscription using GoScript™ Reverse Transcriptase (Promega Corporation, Madison, WI, USA). The product was diluted 3–5 times in TE (10 mM TRIS/1 mM EDTA, pH 8) for subsequent experiments. The cDNA triplicates were amplified via real-time polymerase chain reaction (PCR) in a LightCycler^®^-480 (Roche, Mannheim, Germany) using TaqMan^®^ Fast Advanced Master Mix (Integrated DNA Technologies, IDT, Iowa, USA) and pre-designed TaqMan^®^ probes (IDT, see table primers used for qPCR, A). Actb and Hmbs were tested as housekeeping genes, with Hmbs chosen because of their expression level similarity with the target genes. All the analyzed mRNAs exhibited a Ct value of <25. The relative abundance was corrected using the selected housekeeping gene and the 2−ΔCT algorithm and expressed as arbitrary units.

The miRNAs: Isolated miRNA (4 μL) from kidney and urine samples underwent reverse transcription using the universal cDNA synthesis kit II (Exiqon, Vedbaek, Denmark). Synthetic UniSp6 served as the spike-in control. Incubation was performed at 42 °C for 60 min, followed by heat inactivation at 95 °C for 5 min. The cDNA samples were diluted 60-fold, and 4 μL was used in individual 10 μL PCR reactions with ExiLENT SYBR Green Master Mix and LNA-enhanced miRNA primer assays alongside UniSp6. The real-time quantitative PCR of the circulating miRNAs was conducted as triplicates under the following conditions: 95 °C for 10 min, 40 cycles of denaturation (95 °C, 10 s), annealing (60 °C, 60 s), and melting curve analysis.

Sets of 10 (kidney) and 12 (urine) circulating miRNAs were selected based on the miR-Seq results, including DE genes among the groups. Data normalization against UniSp6 reduced the technical variation, calculated as normalized Ct = Ct of UniSp6—Ct of measured miRNA. NormFinder identified stable miRNAs within normalized smallRNA-Seq data for endogenous controls, selecting miR-181b-5p and miR-411-5p (kidney) and miR-204-5p and miR-181b-5p (urine). The Ct values were calculated using the second-derivative method and reported as arbitrary units.

### 4.11. Statistical Analysis

Descriptive analyses summarized qualitative and quantitative characteristics as percentages and mean values plus standard errors (or median values and interquartile ranges), respectively. The main characteristics between the study and control groups were compared using an unpaired *t*-test or χ^2^ test, as appropriate. The *t*-test was applied to non-parametric data after log transformation. Fisher’s exact test for meta-analysis with Benjamini–Hochberg’s FDR correction was used to calculate the significant targeted biological processes and KEGG pathways. The complete linkage clustering method was used for the hierarchical clustering of pathways and miRNAs, with squared Euclidean distances as distance measures. Absolute P-values were used in all the calculations, considering the significance levels of the interaction. A two-tailed *p*-value of ≤0.05 denoted statistical significance. Statistical analyses were performed using SPSS (version 15.0; SPSS, Chicago, IL, USA).

### 4.12. Study Approval

This study strictly adhered to the Guide for the Care and Use of Laboratory Animals of the Generalitat de Catalunya, Catalan Government. The animal procedures were approved by the Animal Experimentation Ethics Committee of the Centre de Medicina Comparativa i Bioimatge de Catalunya and Germans Trias i Pujol’s Research Institute and Generalitat de Catalunya (CEA-OH/10528/2), following the principles of the Declaration of Helsinki for animal experimental investigations. All the protocols followed the 3R principles and prioritized animal welfare.

## Figures and Tables

**Figure 1 ijms-25-06437-f001:**
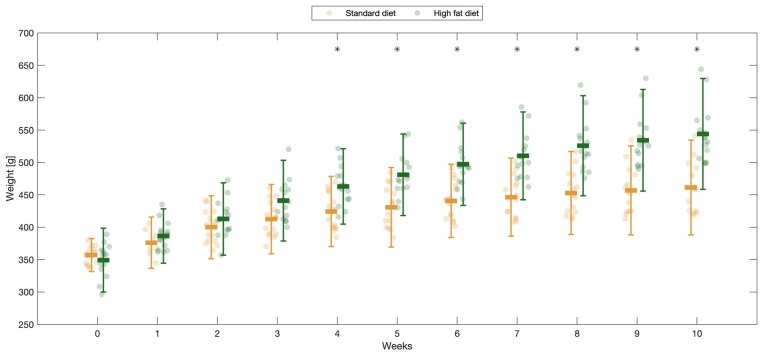
Comparison of rats’ weight evolution during the 10-week follow-up. The standard-diet group is represented in orange, and the high-fat-diet group in green; * *p* < 0.05.

**Figure 2 ijms-25-06437-f002:**
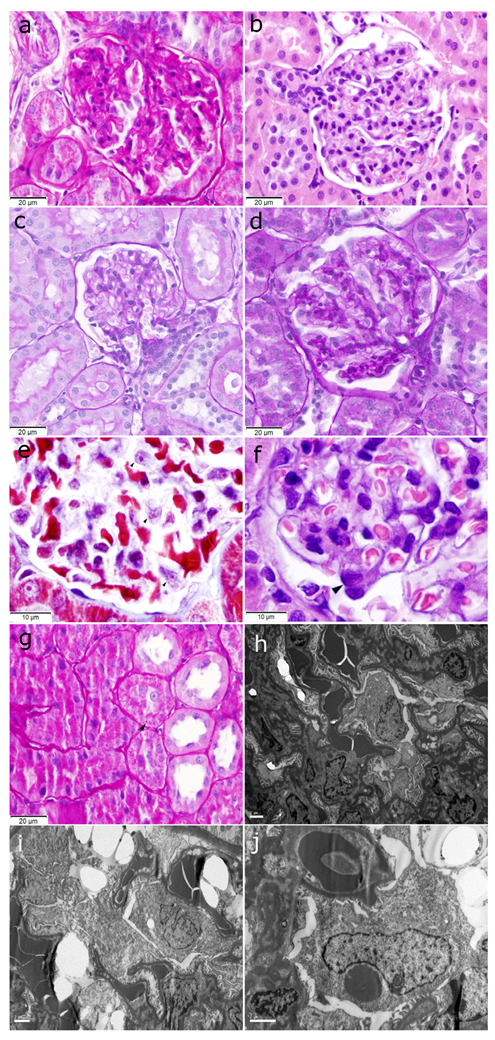
(**a**) Glomerulus with global mesangial matrix increase (PAS stain, original magnification ×400). (**b**) Glomerulus with mesangial cell proliferation (H&E stain, original magnification ×400). (**c**) Tuft glomerular area from a rat fed an SD (PAS stain, original magnification ×400). (**d**) Enlarged tuft glomerular area from a rat fed an HFD (PAS stain, original magnification ×400). (**e**) Podocyte hypertrophy with increased cytoplasm and prominent nucleolus (black arrows) (Masson trichrome stain, original magnification ×1000). (**f**) Podocyte hypertrophy with nuclear enlargement (black arrow) (H&E stain, original magnification ×1000). (**g**) Increased intracytoplasmic resorption droplets (black arrow) in proximal tubular cells from an obese rat fed an HFD (PAS stain, original magnification ×400). (**h**) Transmission electron microscopy image showing swelling of the podocytes, which are hyperplastic, occupying the urinary space (uranyl acetate and lead citrate stain, original magnification ×5000). (**i**) Podocyte cytoplasm showing empty spaces compatible with lipid vacuoles (uranyl acetate and lead citrate stain, original magnification ×6000). (**j**) Podocyte showing a slight increase in cytoplasmic organelles, especially megamitochondria (uranyl acetate and lead citrate stain, original magnification ×10,000). PAS: periodic acid–Schiff; H&E: hematoxylin and eosin; SD: standard diet; HFD: high-fat diet.

**Figure 3 ijms-25-06437-f003:**
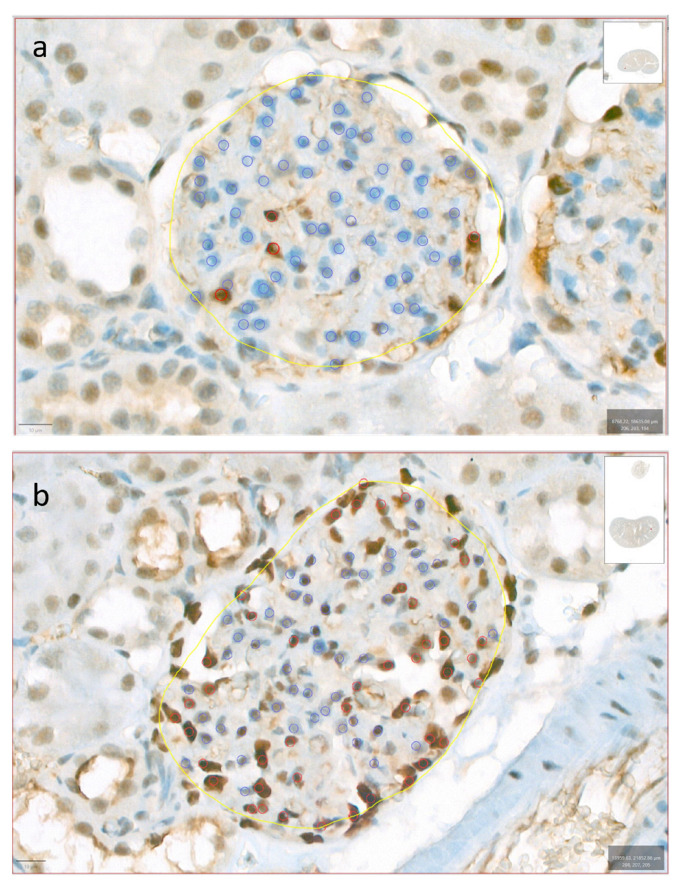
Podocyte densities. This figure shows two distinct examples illustrating low (**a**) and high (**b**) podocyte densities, assessed through immunohistochemical staining for WT1 and quantified using QuPath. Yellow annotations were used to delineate the glomerular region of interest, where the ‘Positive cell detection’ tool was applied. Blue annotations indicate the count of the negative cells, while red annotations highlight positive cells.

**Figure 4 ijms-25-06437-f004:**
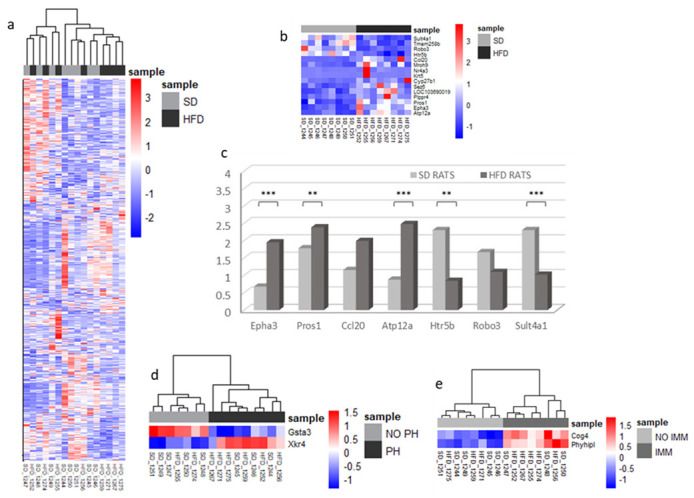
Transcriptomic changes and validation of differentially expressed RNAs. (**a**–**c**) Samples were studied sorted by diet. (**a**) The heatmap displays the expression profiles of all the sequenced genes between the SD-fed and HFD-fed groups. Each row represents a gene, and each column represents a sample. Red indicates upregulation, while blue indicates downregulation. Gene names have been omitted for clarity. Hierarchical clustering was used to group genes and samples based on expression similarity, highlighting similar expression patterns between the two groups. (**b**) Heatmap displaying differentially expressed genes between groups, with red indicating upregulation and blue indicating downregulation. These genes were selected for further analysis; the key differentially expressed genes are listed in Appendix A. (**c**) Validation of the differentially expressed genes in 30 rat kidneys (15 SD vs. 15 HFD) by qPCR. Results are calculated as indices and given in arbitrary units. HFD-rat values are shown in dark-gray bars; SD-rat values in light-gray bars. (**d**,**e**) Heatmaps of differentially expressed genes based on kidney lesions. ** indicates *p* < 0.01; *** indicates *p* < 0.001, unpaired Student’s *t*-test. PH: podocyte hypertrophy; IMM: increased mesangial matrix; SD: standard diet; HFD: high-fat diet.

**Figure 5 ijms-25-06437-f005:**
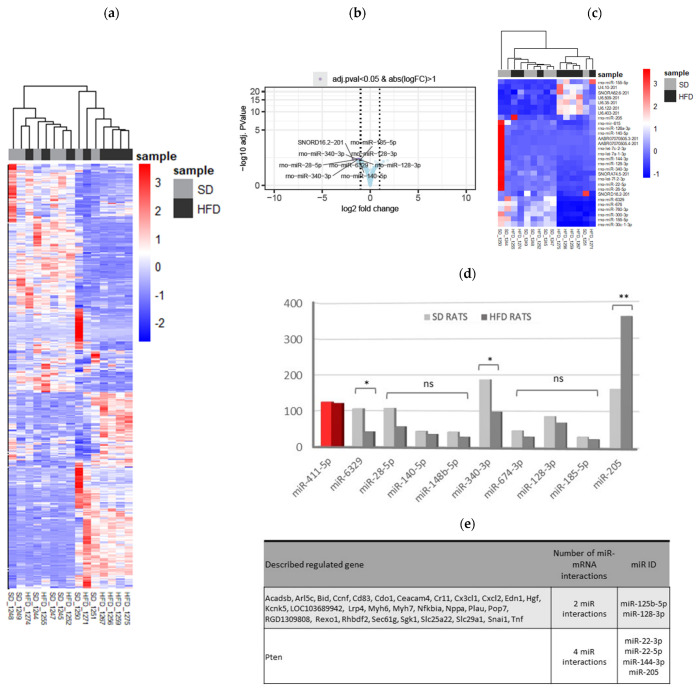
Transcriptomic changes and validation of differentially expressed miRNAs in kidney tissue. (**a**–**d**) Samples were studied sorted by diet. (**a**) The heatmap displays the expression profiles of all the sequenced smallRNAs between the SD-fed and HFD-fed groups. Each row represents a smallRNA, and each column represents a sample. Red indicates upregulation, while blue indicates downregulation. SmallRNA names have been omitted for clarity. Hierarchical clustering was used to group smallRNAs and samples based on expression similarity, highlighting distinct expression patterns between the two groups (**b**) Volcano plot showing that samples in both groups showed a very similar gene expression profile, but some downregulated smallRNA can be observed in the HFD group. (**c**) Heatmap displaying differentially expressed smallRNAs between groups, with red indicating upregulation and blue indicating downregulation. The key differentially expressed smallRNAs are listed in Appendix A. Some miRNAs were selected for subsequent studies. (**d**) The qPCR validation of miRNAs found to be differentially expressed in miRNA-Seq experiments; miR-411-5p was included as an miRNA that does not change its expression (in red). HFD-rat values are shown in dark-gray bars; SD-rat values in light-gray bars; * *p* < 0.05 and ** *p* < 0.001, unpaired Student’s *t*-test. (**e**) Table categorizing validated miRNA-mRNA interactions based on involvement with multiple miRNAs. SD: standard diet; HFD: high-fat diet; miRNA: microRNA; qPCR: quantitative polymerase chain reaction.

**Figure 6 ijms-25-06437-f006:**
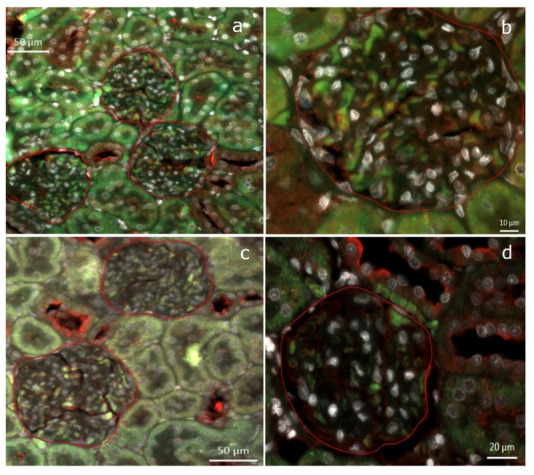
Double immunofluorescence staining for PTEN (green) and nephrin (red) was performed on kidney tissues from an SD-fed rat (**a**,**b**) and an HFD-fed rat (**c**,**d**). The images depict a higher PTEN/nephrin ratio in the SD-fed rat (1.169 in the SD group vs. 0.915 in the HFD group).

**Figure 7 ijms-25-06437-f007:**
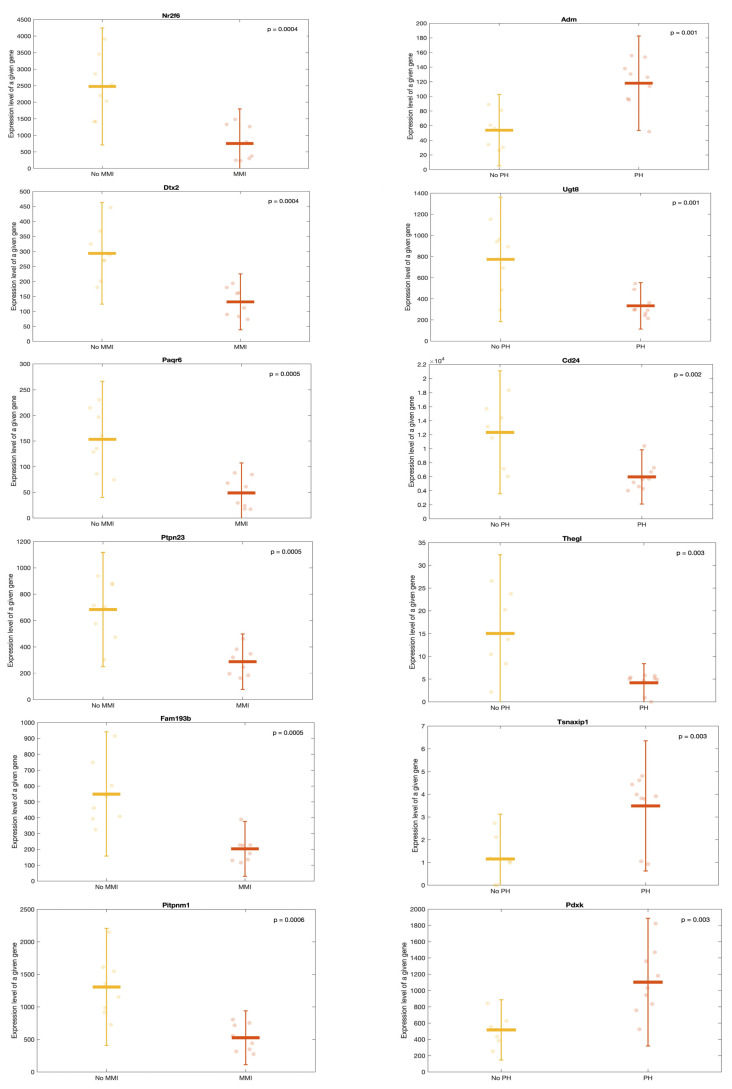
Transcriptomic changes in lesion-related miRNAs in kidney tissue. Scatter plots showing expressions of the first 6 genes correlating significantly with MMI (left) and PH (right). The figures are classified in ascending order of *p*_values (The gene on the top left being the gene with the lowest *p*_value). The points correspond to each individual datum. MMI: mesangial matrix increase; PH: podocyte hypertrophy.

**Figure 8 ijms-25-06437-f008:**
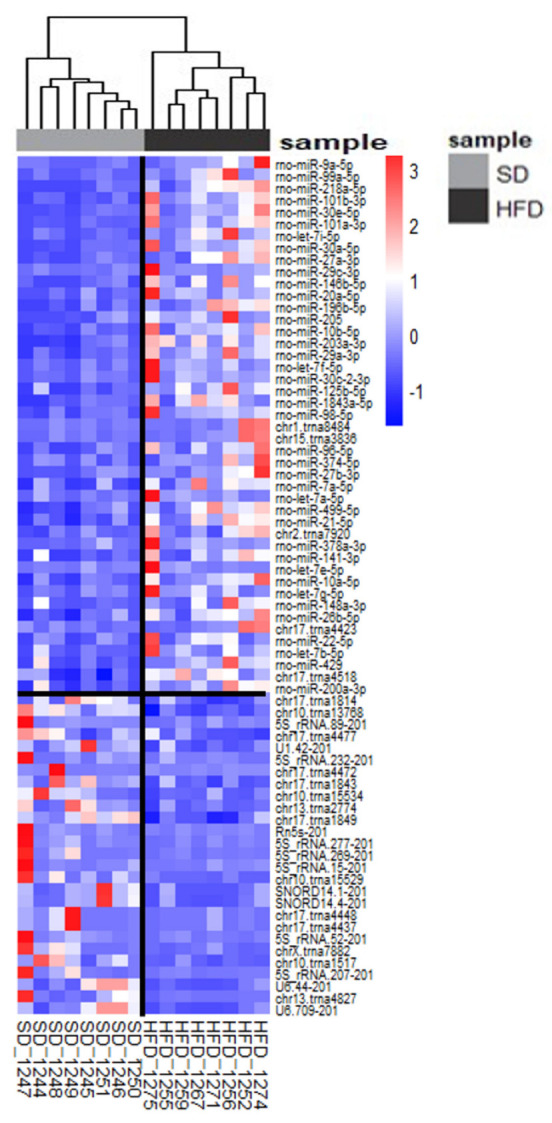
Heatmap representing transcriptomic changes in differentially expressed miRNAs in urine samples sorted by diet. The heatmap displays the expression profiles of differentially expressed smallRNAs between the SD-fed and HFD-fed groups. Each row represents a smallRNA, and each column represents a sample. Red indicates upregulation, while blue indicates downregulation. The key differentially expressed smallRNAs are listed in Appendix A. Hierarchical clustering was used to group smallRNAs and samples based on expression similarity, highlighting distinct expression patterns between the two groups.

**Figure 9 ijms-25-06437-f009:**
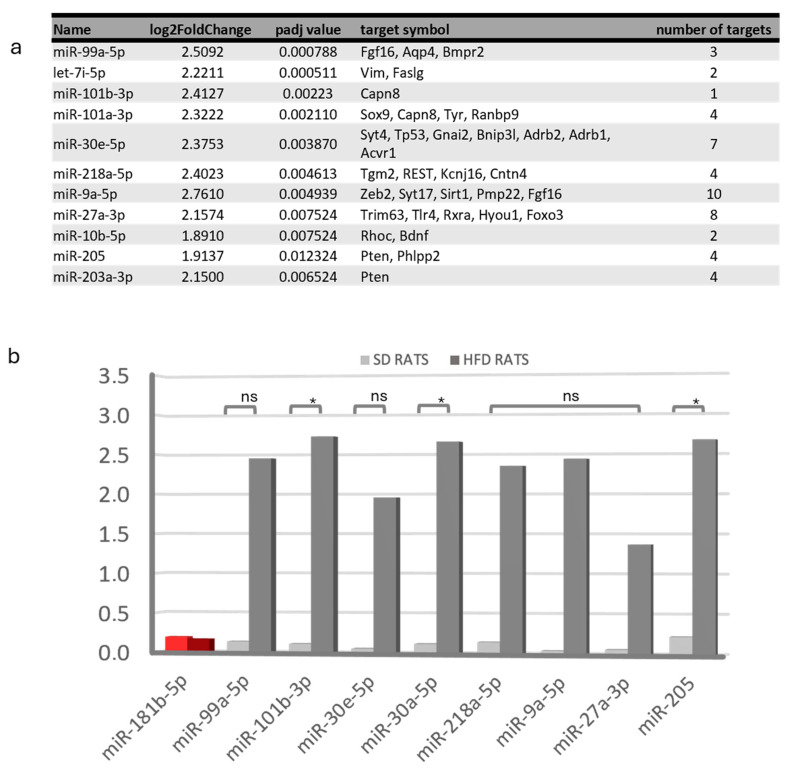
Transcriptomic changes and validation of differentially expressed miRNAs in urine samples sorted by diet. (**a**) Table of the main validated interactions of miRNA-mRNA reported. (**b**) The qPCR validation of selected miRNAs found to be differentially expressed in miR-Seq experiments; miR-181b-5p was included as an miRNA that does not change its expression (in red). HFD-rat values are shown in dark-gray bars; SD-rat values in light-gray bars; * *p* < 0.05, unpaired Student’s *t*-test; “ns”: non-significant. SD: standard diet; HFD: high-fat diet; qPCR: quantitative polymerase chain reaction.

**Figure 10 ijms-25-06437-f010:**
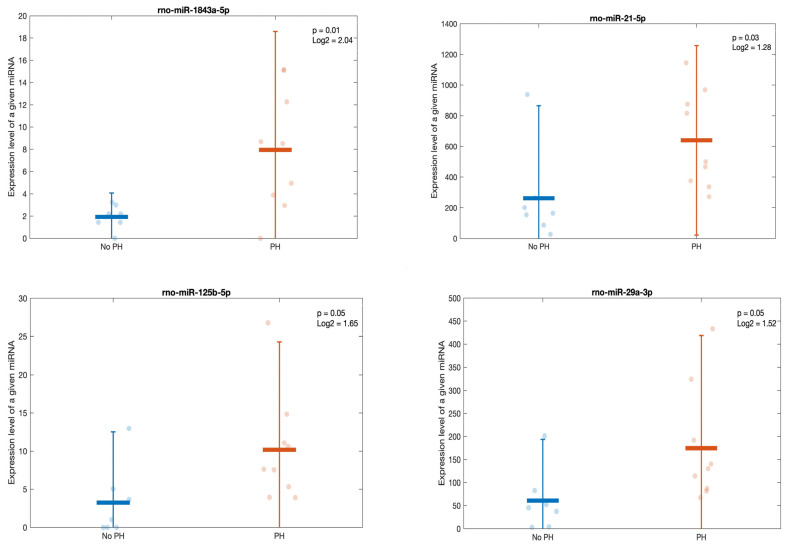
Scatter plots comparing the expressions of the most differentially expressed miRNAs between rats exhibiting podocyte hypertrophy (PH) and rats without this lesion.

## Data Availability

The transcriptomic data supporting the findings of this study are available in the repository Gene Expression Omnibus (GEO) at https://www.ncbi.nlm.nih.gov/geo/ (accession number: GSE262503).

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
