# Peer review of "Integrated miRNA–mRNA Analysis Reveals Critical miRNAs and Targets in Diet-Induced Obesity-Related Glomerulopathy"

_ijms, 2024, doi:10.3390/ijms25126437_

Round 1

Reviewer 1 Report

Comments and Suggestions for Authors

Dear Authors,

I was pleased to be invited to review the article titled Integrated miRNA–mRNA analysis reveals critical miRNAs and targets in diet-induced obesity-related glomerulopathy, a rather informative and insightful original research article aimed at

investigating obesity-related glomerulopathy (ORG) at cellular, structural, and transcriptomic levels.  

The article is praiseworthy in that it appears to be considerably well-conceived, coherently structured (although seeing the Materials and Methods section at the end of the manuscript is quite unusual) and has considerably substantial qualities and strengths: originality, relevance in a highly meaningful area of innovative prognostics research, hence its potential appeal to a relatively wide readership; in addition, it is remarkably comprehensive and thorough in terms of pursuing its stated objective. It relies on sound and well-explicated methodology, as far as I was able to determine, and the tables are also well-crafted and effective in conveying key data and findings. 

Having said that, I feel that there are still several shortcomings which need to be addressed: 

- the article's objective is not stated clearly or thoroughly enough. Please outline more clearly what you mean to illustrate and emphasize with more clarity why and to what extent it is significant to the research community. Expounding upon such key aspects more thoroughly would help flesh out your objective in a more comprehensive and better defined fashion. Starting with the abstract, such a shortcoming is glaring. The potential value and applicability of the study's findings are not well explicated. That detracts from the study's appeal.

- By the same token, it is advisable to provide a higher degree of contextualization and elaboration on the significance of the study's findings in terms of dealing with the novel potential approaches arising from personalized/precision medicine, molecular classifications, ncRnas. Such highly innovative and still evolving avenues for diagnostics and prognostics have been harnessed in the prognostic and therapeutic approaches for various conditions, e.g. cancer research, hence a brief comparative analysis would certainly benefit the scope of the article as a whole.

The Discussion is in fact the part which falls relatively short compared to the level of thoroughness and accuracy which the Authors were able to achieve while pursuing the article's chief objective. 

Furthermore, a much more in-depth analysis ought to be provided as to the tailored approaches arising from personalized/precision medicine-based techniques, also in terms of their implications for policy-making and evidence-based guidelines, as it is the case in many countries. The issue of consent, which presents unique complexities given the magnitude of such innovations, should also be briefly elaborated on, in light of the unique complexities arisisng from data processing, for instance, and novel AI-based approaches to care. 

Such elements of discussion are likely to have a bearing on possible malpractice claims if the most suitable therapeutic avenues are not pursued. When you mention that renal biopsy in the early stages of ORG is not acceptable in clinical practice or investigations for ethical reasons, such a  statement needs further elaboration and a higher degree of contextualization, as does the importance of tailored approaches, or that remark will stay underdeveloped. 

The following sources should be drawn upon and cited:

Cavaliere AF, Perelli F, Zaami S, Piergentili R, Mattei A, Vizzielli G, Scambia G, Straface G, Restaino S, Signore F. Towards Personalized Medicine: Non-Coding RNAs and Endometrial Cancer. Healthcare (Basel). 2021 Jul 30;9(8):965. doi: 10.3390/healthcare9080965.

Legal Ramifications of Ambiguous Clinical Guidelines. JAMA. 2017 May 16;317(19):2020. doi: 10.1001/jama.2017.4501.). 

While the article is coherently crafted and relatively well written overall some adjustments are needed along with a broader scope, in order to clarify the applicability and impact of its findings for ORG prognostics/therapeutics and beyond, and for public health as a whole. Thus, by making the most out of its premise and primary objective, the article could make for a valuable contribution to a uniquely meaningful research area and be attractive to a relatively broad readership. Further proofreading by a native speaker of English is recommended. 

Best regards.

Comments on the Quality of English Language

Well-flowing and effectively enunciated overall. Further proofreading by a native speaker of English is still recommended. 

Author Response

Pllease see the attachment

Reviewer 2 Report

Comments and Suggestions for Authors

In this paper, the authors report the results of a project aimed at the description, by means of biological / histological, high-throughput and bioinformatic analysis, of obesity-related glomerulopathy in a rat model. 

The authors used 30 Wistar rats randomized into two groups: control (n=15) and study groups (n=15) that were fed with a standard (SD) and high-fat diet (HFD). After 10 weeks, a set of investigations was performed in order to understand differences between the two groups. Weight, kidney function, histological features, urine, transcriptomic changes for mRNA as well as microRNA genes were investigated. From a molecular point of view, the authors focus their attention on mRNA – miRNA relationships. The eventually stress the role of the PTEN gene.

My opinion on this manuscript is that it potentially represents an interesting project, designed to recover a broad array of data concerning the animal model and the biological issue under study. The large set of RNA-seq data reported can be of interest for scholars working in the field.

However, there are two major issues here:

- the apparent overall scarcity of the results from the transcriptomic analysis which raises questions about the project design.

- the presentation and discussion of the data, which are quite unclear and riddled with inaccuracies all along the current version of the manuscript.

Therefore, I have these major comments:

1) There is a general issue with the quality of the figures. In the PDF version of the manuscript I downloaded, all the figures are of very low resolution and in some cases, details are not readable.

2) As I mentioned before, this project in principle collected a lot of high throughput data. However, results seem to be minimal, at least in term of differentially expressed genes and miRNA. Why this happens ? Could the authors comment on that? Is this a problem in study design, sample collection, variability in the data ?

3) In the following, I’m reporting a set of sentences and points in the manuscript that all must be modified because incorrectness or not fully professional.

- To profile diet-related gene changes in the kidneys, RNA-Seq experiments were performed …. detecting 14,700 sequences.

I didn’t understand this point … what does it mean "sequences" ? Maybe the authors mean “14,700 expressed genes” ?

- A biological significance analysis identified overrepresented pathways, including those related to renal injury prevention (Epha3 and Pros1), inflammation (CCL20), and nervous system function (Sult4a1, Robo3, and Htr5b)

This is incorrect. Starting from a set of only 5 or 6 genes, it is impossible to discuss overrepresented pathways. With so few genes, the only feasible approach is to annotate each gene to its individual functions.

- In the heatmap pf Figure 4a, it is not useful to display the gene names, since they are overall absolutely not readable.

- In the legend of Figure 4b: “Heatmap of over and underrepresented groups of genes (red and blue respectively) that clearly discriminate samples among groups. Some of these genes were selected for subsequent studies. Results are given as number of reads per gene”

This sentence is wrong and / or not professional. What does it mean “over and underrepresented groups of genes” ? What does it mean: “ Results are given as number of reads per gene” ? The number of raw reads per gene cannot be used for gene quantification, whilst FPKMs / RPKMs / TPMs are usually used. Moreover, the color code bar displayed at the right of the heatmap apparently counts also negative number, therefore this cannot be a raw reads number. Heatmaps are usually produced in a manner that each row is centered around the mean or median of each gene, like a Z-score, that could be negative.

- "Hierarchical clustering and principal component analysis" 

Please note that there is no principal component analysis plot all along the paper.

- "A total of 812 sequences were detected”.

See my previous comments, maybe the authors mean “812 small-RNA genes were detected” ?

- Thirteen miRNAs were identified using miRTarBase and TargetScan, with nine DE genes (miR-205, miR-140-5p, miR-185-5p, miR-128-3p, miR-28-5p, miR-340-3p, miR-6329, miR-674-3p, and miR-148b-5p)

The meaning of this sentence is unclear to me. In the previous line, the authors state that 338 miRNA where detected and 15 where found differentially expressed. What happened to the 2 miRNA not further considered ?

- GO analysis revealed targets associated with inflammatory responses; tissue injury and repair; and nervous tissue, fat, and mitochondrial functions.

Exactly which set of genes does this statement refer to? Why you don’t’ have genes associate also to a combination of 3 different database ?

- Heatmap of over- and underrepresented groups of interesting genes.

See my previous comment.

- Are the genes reported in Table 1 expressed in the mouse model used ? 

- Moreover, note that the use log2FoldChange > 1, is very strange. Is it correct, so that only positively regulated genes where used in the subsequent analysis? Maybe the authors refer to absolute log2FoldChange > 1 ?

- Paragraph 2.6 and also region 2.8: should be completely rewritten. The usage of simulations (random datasets) is really unclear.

- In paragraph 2.9 I think that log2FoldChange>1 is not correct, since in the heatmap are showed but upregulated as well as downmodulated genes (see my previous comments).

- In Table 9a, it is unclear where the “validated interactions miRNA-mRNA” reported actually come from ? 

- Is PTEN actually modulated in the RNA-seq ?

- In the methods section, the authors state that: “Gene set enrichment analysis was performed using gene ontology (GO) and the Reactome Pathway Knowledgebase for biological significance analysis” and they also cite REVIGO. But where Reactome Pathway Knowledgebase plus REVIGO was actually used ? 

- I suggest the authors add a set of Supplementary files (Tables), in which  the RNA-seq data are directly reported, without the need of access the GSE.

Round 2

Reviewer 1 Report

Comments and Suggestions for Authors

Dear Authors,

I can certainly appreciate the level of depth and elaboration you have attained in this revised version. Further expanding on miRNA-based diagnostic/therapeutic avenues and adding more breadth and contextualization to patient care/stratification have made the article more comprehensive and well-rounded overall.

I am therefore recommending it be approved for publication.

Best regards.

Reviewer 2 Report

Comments and Suggestions for Authors

Dear editors,

I think that in this revised version the authors meet all the previous concerns adeguately.

The exact explanation for the low number of modulated genes found remains partially unclear; however the dataset and the reported analysis can be useful for readers interested in the topic.